# COVID-19 Biogenesis and Intracellular Transport

**DOI:** 10.3390/ijms24054523

**Published:** 2023-02-24

**Authors:** Alexander A. Mironov, Maksim A. Savin, Galina V. Beznoussenko

**Affiliations:** 1Department of Cell Biology, IFOM ETS—The AIRC Institute of Molecular Oncology, Via Adamello, 16, 20139 Milan, Italy; 2The Department for Welding Production and Technology of Constructional Materials, Perm National Research Polytechnic University, Komsomolsky Prospekt, 29, 614990 Perm, Russia

**Keywords:** COVID-19, SARS-CoV-2, Golgi complex, virion budding, endocytosis, viral replication, intracellular transport, viral replication organelle

## Abstract

SARS-CoV-2 is responsible for the COVID-19 pandemic. The structure of SARS-CoV-2 and most of its proteins of have been deciphered. SARS-CoV-2 enters cells through the endocytic pathway and perforates the endosomes’ membranes, and its (+) RNA appears in the cytosol. Then, SARS-CoV-2 starts to use the protein machines of host cells and their membranes for its biogenesis. SARS-CoV-2 generates a replication organelle in the reticulo-vesicular network of the zippered endoplasmic reticulum and double membrane vesicles. Then, viral proteins start to oligomerize and are subjected to budding within the ER exit sites, and its virions are passed through the Golgi complex, where the proteins are subjected to glycosylation and appear in post-Golgi carriers. After their fusion with the plasma membrane, glycosylated virions are secreted into the lumen of airways or (seemingly rarely) into the space between epithelial cells. This review focuses on the biology of SARS-CoV-2’s interactions with cells and its transport within cells. Our analysis revealed a significant number of unclear points related to intracellular transport in SARS-CoV-2-infected cells.

## 1. Introduction

The first coronavirus was discovered in 1965 [1]. Now, there are seven members of the coronavirus category, which induce dangerous diseases in human beings [2]. Virions of SARS-CoV-2 are spherical. Their diameter is equal to 90–100 nm. On the surface of this virus there are 48 spikes or peplomers, with their length being equal to 23 nm. However, the spike number on each SARS-CoV-2 viral particle varies according to different sources in the literature. So, it should be considered as a range of about 25–50 [3,4,5,6,7,8,9,10,11,12]. The ultrastructures of SARS-CoV-2 and SARS-CoV are similar [3,4,5,6,7,8,9,10,11,12,13,14]. However, the number of peplomers is lower in SARS-CoV-2 [9]. The S-protein of SARS-CoV-2 differs by the presence of the unusual furin cleavage site situated between subunits of the S-protein. SARS-CoV-2 contains only one positive-sense RNA, and only its single chain. Its 5′ end exhibits a methylated cap, whereas its 3′ end contains several adenines. [15,16,17,18,19,20]. Also, there is a difference of one amino acid between the C-tails of spikes from SARS-CoV-2 and SARS-CoV: Cys1247 vs. Ala1229, respectively [21].

The RNA and nucleoproteins are surrounded with a lipid bilayer composed of phospholipids. Its thickness is lower (3.6 ± 0.5 nm) than that of the membranes of the host cell (3.9 ± 0.5 nm) because this bilayer is the derived from the ER membrane (see below). Among proteins synthesized on the basis of the RNA of SARS-CoV-2, there is a papain-like protease. The RNA is associated with the viral matrix proteins. Host ribosomes translate a large polypeptide using two-thirds of the RNA [4,5,22,23,24].

Being experts in intracellular transport [25,26,27,28], we tried to examine the problem of SARS-CoV-2 biogenesis from the point of view of intracellular transport. We used the available information on intracellular transport and in particular transport through the GC, with that describing virus trafficking, and tried to predict several important aspects of the interactions between cells and SARS-CoV-2 virions.

## 2. Proteins of SARS-CoV-2

The genome of SARS-CoV-2 (30 kb) is one of the largest among RNA viruses. The SARS-CoV-2 genome is composed of a single-stranded RNA of ~29,900 base pairs with a methylated cap at the 5′ end and a polyadenylated (poly-A) tail at the 3′ end [10]. It encodes 29 proteins (Table 1), including 4 structural proteins (Spike (S), Envelope (E), Membrane (M), and Nucleocapsid (N)), 16 non-structural proteins (NSP1–NSP16), and 9 accessory factors (ORF3a, ORF3b, ORF6, ORF7a, ORF7b, ORF8, ORF9b, ORF9c, and ORF10) [29,30,31]. The structural proteins are encoded by the one-third of the genome near the 3′-terminus [5,12,22]. The S, E, and M proteins are membrane-associated proteins, localized to the ER and GC. They interact with N proteins and drive the assembly of new virus particles, ensuring vRNP incorporation into the nascent virion [29,30,31,32]. The single-chain positive sense RNA of SARS-CoV-2 is used directly as mRNA [18]. Using this mRNA, ribosomes of the host cells synthesize two large polypeptides, rep1a and rep1a/1b. Then, rep1a and rep1b are proteolytically cleaved into 16 virally-encoded nonstructural proteins (NSPs) [33,34,35,36,37,38].

The spike S-protein is a 600 kDa trimeric transmembrane type I membrane glycoprotein that is composed of two subunits, S1 and S2 [31]. It forms peplomers [17,39,40,41]. The S-proteins of SARS-CoV and SARS-CoV-2 share 75% identity in amino acid sequences [18,19,20]. In the S-protein of SARS-CoV-2 there is a distinct four-amino-acid insert located at the interface between the S1 receptor-binding subunit and the S2 fusion subunit. These structural features of SARS-CoV-2 RBD increase its ACE2 binding affinity [42]. 

The high degree of glycosylation of the S-protein indicates that viruses pass through the GC. The S-protein binds to ACE2 through the receptor-binding domain [43,44,45,46]. Its 66 N-linked glycans are formed in the GC and contain a sialic acid-binding pocket that can mediate viral attachment by interactions with various sialoproteins, glycoproteins, and gangliosides on the cell membrane [47,48,49,50,51]. The S1 subunit is responsible for host cell receptor binding, while the S2 subunit participates in membrane fusion. The modified glycans shield about 40% of the protein surface of the S trimer. Most of glycans on the surface of the S-protein have sialic acid on their termini [31]. SARS-CoV-2 is decorated by a large number of highly glycosylated proteins, and its glycosylation (both N-linked and O-linked) extensively affects host recognition, penetration, binding, recycling, and pathogenesis. The extensive N-glycosylation of the SARS-CoV-2 spike protein could cause the negative effects of the virus interacting with the host cell ACE2 receptor [31].

Each subunit contains a single transmembrane domain, an N-terminal domain, and a C-terminal domain [17,18,19,20,21,41,44,45,46,51]. The fusogenic S2 subunit consists of the upstream helix region, the fusion peptide, the heptad repeat 1, the central domain, the heptad repeat 2, the transmembrane domain, and the cytoplasmic tail [51]. The cytoplasmic tail of the S-protein contains a di-lysine ER-retrieval signal, which interacts with COPI [41,52,53,54]. This ER retrieval motif is present in both SARS-CoV and SARS-CoV-2 [7,8,9,10]. It interacts with the whole COPI coatomer complex [21]. The S-protein interacts with the host Rab7A, Rab7B, and Rab7L1, as well as with VPS11 and VPS33A [55,56]. 

After binding of the S-protein subunits, the S1/S2 complex is subjected to cleavage by the human type II transmembrane serine protease 2 (TMPRSS2), dipeptidylpeptidase 4 (DPP4), and furin. This facilitates viral entry into the cytosol [23,44,53,54,55,56,57,58]. The SARS-CoV-2 S-protein may be more readily primed for membrane fusion than that of SARS-CoV because the latter requires two proteolytic events after receptor binding. [51]. The S-protein is cleaved at the multi-basic site into S1 and S2 subunits by either furin at the trans-Golgi network (TGN) or by another proprotein convertase [21]. During initial processing, after S1–S2 cleavage, the spike then either recycles back to the ER–Golgi intermediate compartment (ERGIC) for assembly into SARS-CoV-2 virions or traffics to the plasma membrane [21]. After its cleavage, the S-protein pierces the double paired membrane composed of the PM and viral membrane, and the nucleocapsid penetrates into the cytosol. 

On the other hand, binding of the S1 subunit to the ACE2 receptor triggers the cleavage of ACE2 by ADAM17/tumor necrosis factor-converting enzyme at the ectodomain sites. A soluble form of ACE2 is produced. It retains its catalytic activity [57]. A polybasic residue motif at the boundary between the S1 and S2 subunits is cleaved by furin and furin-like proteases during biogenesis and cell entry [16,43,51].

The 25 kDa type III transmembrane glycoprotein M binds to the nucleocapsid and favors the curvature of the host cell membrane. This triple-spanning membrane (M) protein is the most abundant membrane protein component of the viral envelope. The M protein exists as monomers in the ER, but it oligomerizes to form variously sized complexes during transport through the GC and trans-Golgi network (TGN). The cytoplasmic domains of M proteins homo-oligomerize. This contributes to its retention. The M protein and E protein do not increase the thickness of the viral membrane envelope [58,59,60].

The E-protein encoded in the genome of an RNA virus is crucial for the replication, budding, and pathophysiology of the virus. During virus budding, the E-protein is not specifically targeted at the membranes of ER exit sites but displays a broader distribution in the Golgi region. The N-protein forms biomolecular condensates with RNA. The SARS-CoV-2 envelope protein forms clustered pentamers in lipid bilayers. The M–E complex ensures the uniform size of viral particles for viral maturation and mediates virion release. It is not clear if polysaccharide chains on the M- and E-proteins exist. If they exist, it is not clear how glycosylation of these proteins and their transport through the GC occurs: in the form of a dietary supplement or separately [58,59,60].

The N-protein and the majority of the NSP proteins have not been studied intensively (Table 1). The fine structures of the most of SARS-CoV-2 NSP proteins have been deciphered [17,18,33,46,61,62].

**Table 1 ijms-24-04523-t001:** Functions of SARS-CoV-2 proteins.

Name of Protein	Function	Reference
S	The spike S-protein binds to its receptor (ACE2) and perforates (after its cleavage) the double membrane composed of the PM and endosomal membrane.	[31,33,38]
E	The envelope (E) protein is crucial for budding and replication	[52,58,59,63,64,65,66]
M	The triple-spanning membrane M protein binds to the nucleocapsid and generates the curvature of the host cell membrane. The cytoplasmic domains of M proteins homo-oligomerize. The M–E complex ensures the uniform size of viral particles for viral maturation and mediates virion release.	[62]
N	The N protein forms biomolecular condensates with RNA.	[61,63]
NSP1	NSP1 binds to 18S ribosomal RNA in the mRNA entry channel of the ribosome and leads to global inhibition of mRNA translation upon infection. It degrades host mRNA and inhibits IFN signaling. NSP1 suppress IFN-I signaling.	[67,68]
NSP2	NSP2 induces repression of mRNA translation, impairing interferon production.	[69]
NSP3	NSP3/papain-like protease is responsible for cleaving viral polyproteins during replication. NSP3 is the largest protein encoded by the coronavirus. It cleaves viral proteins. NSP3 reverses PARP9/DTX3L-dependent ADP-ribosylation induced by interferon signaling.	[70,71]
NSP4	NSP4 induces pro-inflammatory mitochondrial DNA release in inner membrane-derived vesicles.	[72]
NSP5	NSP5/3C-like protease is responsible for cleaving viral polyproteins during replication. NSP5 exhibits catalytic activity and interferon antagonism. NSP5 activates the NF-κB pathway.	[73,74,75,76]
NSP6	NSP6 participates in the biogenesis of the SARS-CoV-2 replication organelle. It forms double membrane vacuoles (DMVs). NSP6 impairs lysosome acidification to inhibit autophagic flux. NSP6 suppress IFN-I signaling. It triggers NLRP3-dependent pyroptosis.	[14,76,77]
NSP7	They form a dexadecameric complex. They form polymerase complex consisting of the NSP12 catalytic subunit and NSP7–NSP8 cofactors. Processing clamp for RdRp. NSP8 and NSP9 bind to the 7SL RNA in the signal recognition particle and interfere with protein trafficking to the cell membrane upon infection.	[67,78]
NSP8		
NSP9	RNA binding. NSP8 and NSP9 bind to the 7SL RNA in the signal recognition particle and interfere with protein trafficking to the cell membrane upon infection.	[67]
NSP10	SARS-CoV-2 NSP10/NSP14-ExoN and NSP16 functions as an exoribonuclease with NSP14.	[79]
NSP11	Real function is unclear. NSP11 has a helical propensity.	[80]
NSP12	RNA-dependent RNA-cap2′-O-methyltransfeaze. Viral mRNA capping protecting it from RNases and RNA polymerase. Viral replication and transcription. Forms polymerase complex consisting of the NSP12 catalytic subunit and NSP7-NSP8 cofactors. NSP12 is the viral RNA-dependent RNA polymerase (RdRp) and suppresses host antiviral responses.	[78,81]
NSP13	RNA helicase, 5′-triphosphatase. 2 NSP13, NSP14, and NSP15 function as potent interferon antagonists.	[82,83,84,85,86]
NSP14	NSP13, NSP14, and NSP15 function as potent interferon antagonists.	[86,87,88,89]
NSP15	Endoribonuclease NSP13, NSP14, and NSP15 function as potent interferon antagonists.	[86]
NSP16	2′-O-methyltransferase NSP16,	[14,90]

## 3. Cell Structure of Human Airways

For practical purposes, it is especially important to examine the infection of epithelial cells of the airways. It is important to understand how the SARS-CoV-2 penetrates not only into cells in culture, but also into polarized cells lining the respiratory tract. Therefore, we would like to mention that most cases of SARS-CoV-2 infection begin in the upper respiratory tract. The epithelium of the respiratory tract consists of ciliated cells, goblet cells, club cells, and underlying basal cells. The ciliated cells contain not only cilia, but also a significant number of short microvilli. The apical surface of the club cells is covered with microvilli. The fact that ciliated cells also contain cilia and microvilli can be observed in the following figure (http://histologyguide.com/EM-view/EM-077-respiratory-epithelium/17-photo-1.html, http://histologyguide.com/EM-view/EM-076-respiratory-epithelium/17-photo-1.html, http://histologyguide.com/EM-view/EM-070-respiratory-epithelium/17-photo-1.html (accessed on 2 February 2023).

Ciliated cells, together with goblet-shaped and club cells, participate in catching inhaled particles in the secreted mucus. Then, the beating cilia transport it up into the mouth, where it is then swallowed or expectorated. Basal cells are able to differentiate into the other cell types mentioned above. Under normal conditions, mucus and cilia prevent the access of any possible irritants or large pathogens to epithelial cells [91,92,93,94].

Rare lymphoid follicles and single plasma cells producing IgA and are observed in the lamina propria of bronchi. After its secretion from plasma cells localized within the lamina propria, IgA should be delivered into the lumen of the airways. The most plausible candidate for this role are goblet cells, where after basolateral endocytosis IgA is transported towards the GC and then secreted together with mucus. Goblet cells are the main producer of mucus, and it is important to understand their role. The mucus produced by these cells could be protective and prevent much of the virus from accessing the goblet cell surface [35]. Heparan sulfate promotes SARS-CoV-2 infection in various target cells [51,92]. In lungs, DCs were not identified at the EM level [93,94]. Although human airways are not available for experimentation, airway cell-derived organoids that resemble the lungs’ structure and function ex vivo could be used now.

## 4. ACE2 as a Receptor for SARS-CoV-2

The entry of SARS-CoV-2 into cells is based on its immobilization on the cell surface after its interaction with ACE2. Attachment of SARS-CoV-2 to the PM of cells is mediated with ACE2. Without ACE2 SUMOylation, SARS-CoV-2 infection could be blocked [95,96]. A549 cells exhibiting a low level of ACE2 expression are less sensitive to infection with SARS-CoV-2. Their transfection with ACE2 makes them sensitive to SARS-CoV-2 [43,97]. Also, the S-protein binds to Toll-like receptors of pneumocytes [98]. SARS-CoV-2 infection leads to increased activation of MT1-MMP that is colocalized with ACE2 and stimulates cell entry of SARS-CoV-2. Inhibition of MT1-MMP suppresses entry of SARS-CoV-2 into cells [99]. In epithelial cells, being localized on the APM of epithelial cells of airways. ACE2 serves not only as an enzyme but also as a chaperone controlling intestinal amino acid uptake, regulating gut amino acid transport into cells [100].

The SARS-CoV-2 receptor, angiotensin-converting enzyme 2 (ACE2) or ACE homolog (ACEH), is a membrane protein of the first type consisting of 805 amino acids. The full-length metallocarboxyl peptidase angiotensin receptor (mACE2) is located on cell membranes and consists of a transmembrane anchor and an extracellular domain. Its gene is located within the X chromosome. ACE2 has 40% identity and 61% similarity to ACE metalloprotease [101]. It has a signal peptide, an extracellular N-terminal domain containing a conserved catalytically active HEXXH zinc-binding domain, a transmembrane domain (amino acids 740 to 763), and a C-terminal cytosolic tail (mostly in humans) close to the C-terminus (type I membrane protein) [102,103]. Its extracellular domain exhibits monocarboxypeptidase activity (it is a zinc metalloprotease) and binds to the S1 domain of the S-protein [14,15,17,61]. ACE2 converts Ang (angiotensin) I and Ang II into Ang 1–9 and Ang 1–7, respectively, and several other peptides. ACE2 can also act on [des-Arg 937]-bradykinin of the kinin–kallikrein system, regulating coagulation and inflammation [101].

Different proteases can cleave ACE2. The second form of ACE2 is a soluble form detectable in the blood, although rarely [57]. This form of ACE lacks membrane anchors and circulates in low concentrations. The analysis of ACE2 expression in experimental models and in the human transcriptome by using different databases revealed that its level is very low in the lung, being mainly limited to a small fraction of type II alveolar epithelial cells.

ACE2 is normally localized on the plasma membrane (mACE2) with the N-terminal containing the catalytic site protruding into the extracellular space, using as substrates different active peptides present in the interstitial space. ACE2 is found on microvilli but not on the cilia of the APM of epithelial ciliary and club cells of the airways. No specific sorting signals and domains responsible for the apical sorting of ACE2 were found within the S-protein. It is possible that the transfer of ACE2 to APM is associated with an ultra-high level of glycosylation of the extracellular domain of ACE2 of these cells. The presence of a large number of sialic acids on polysaccharides synthesized on ACE2 is associated with its apical sorting. The hydrogen bonds formed at the acid ends polymerize the ACE2 molecules and make rafts out of them, which are then delivered to the APM. In the culture of unpolarized cells, ACE2 is located on BLPM due to the fact that there are domains on this PM that came from endosomes, as derivatives of the APM with a thick membrane [57].

SARS-CoV-2 is delivered to ACE2 localized on microvilli of ciliated or club cells due to the intensive movement of cilia. Motile cilia ensure SARS-CoV-2 delivery to the cell body through mucus layer entry. Depleting cilia inhibits COVID-19 [104]. The cystic fibrosis transmembrane conductance regulator (CFTR) channel regulates the expression and localization of ACE2. Due to impairment of the function of mucus, patients with cystic fibrosis are less sensitive to COVID-19. There is upregulation of ACE2 and TMPRSS2 expression in the airways of patients suffering from cystic fibrosis [105]. How ACE2 is transported and whether the COPII coat and COPII vesicles are involved in this process are not clear. 

## 5. ACE2 Localization

ACE2 is widely expressed in various tissues and organs [78,106,107]. High levels of ACE2 expression are observed in the lungs, brain, kidneys [103,108], small intestine, testis, heart muscle, colon, and thyroid gland. However, no ACE2 is detected in blood cells [57,101]. SARS-CoV-2 was not detected on ciliary membranes but observed on the microvilli of ciliated cells. SARS-CoV-2 is associated with microvilli and the apical plasma membrane. ACE2 is not detected on the APM of goblet epithelial cells. Also, the apical plasma membrane of goblet cells does not contain endocytosis machinery suitable for the viral entry [35].

Labeling for ACE2 is visible in the brush border (microvilli) of differentiated enterocytes in the ileum, duodenum, jejunum, caecum and colon [109,110]. ACE2 is expressed in organoids composed of enterocytes. ACE2 was also present in endothelial cells from small and large arteries and veins in all the tissues studied. SARS-CoV-2 infects endothelial cells because they contain ACE2 [18,35,110,111,112,113,114]. It is unclear why labeling for ACE2 is present only in some cells, whereas in situ labeling for APM, i.e., in enterocytes, is extremely uniform [114,115,116,117,118,119]. 

In the olfactory mucosa, ACE2 is highly expressed on microvilli of the sustentacular cells, which form an epithelial monolayer around the olfactory neuronal outgrowths (bulbs) penetrating through this monolayer of cells and ending in olfactory pins. ACE2 is not detected on the olfactory sensory neurons themselves (olfactory neuronal bulbs) [111,120]. Thus, it seems that SARS-CoV-2 is not a neurotropic virus [110,120,121]. The problems with taste and smell in COVID-19 patients are not related to SARS-CoV-2 entry into neurons [111,122]. On the other hand, it is proposed that SARS-CoV-2 enters the central nervous system using different pathways [14]. Indeed, 7 days after cell entry, SARS-CoV-2 was observed in the olfactory cortex in the brains of rhesus monkeys [123]. Also, in patients who died with COVID-19, SARS-CoV-2 is demonstrated in several respiratory and non-respiratory tissues, including the brain [124,125].

ACE2 expression exhibits a gradient from upper to lower airways [109,110,111,113,114,126]. ACE2 is not present within goblet cells (secretory goblet cells make up ~20% of the epithelial cells) of the upper and lower respiratory tract [127]. MUC5B+ “club” and MUC5AC+ goblet cells were not infected in vivo [18]. The mucus produced protects goblet cells [35]. ACE2 is observed in the duct-lining epithelial cells and acinar cells of major salivary glands [109,114]. There is no evidence of virus entry into alveolar macrophages, whereas salivary glands are a target for SARS-CoV-2 [109]. Type II pneumocytes taken from dyed healthy lungs of non-human primates contain ACE2 (see Figure 1B of [15]). Expression of ACE2 and the above-mentioned accessory proteases is higher in males and increases with age [128,129,130,131]. SARS-CoV-2 infection depends also on cellular heparan sulfate, which changes the spike structure to an open conformation to facilitate ACE2 binding [132].

Children express less ACE2. As a result, they are more resistant to SARS-CoV-2 [128]. An increased level of soluble ACE2 correlates with disease severity. ACE2 expression is stimulated by a type I interferon gene in human airway epithelial cells [36,57]. IFN stimulation upon viral treatment induces the expression of a truncated ACE2 isoform, which lacks 356 N-terminal amino acids, is not able to bind SARS-CoV-2, and, therefore, does not contribute to the potentiation of the infection [57].

## 6. Role of TMPRSS2

TMPRSS2 is the human type II transmembrane serine protease able to promote ACE2 proteolytic cleavage using different targets in the protein sequence. It cleaves ACE2 at the intracellular C-terminal domain and, differently from ADAM17 does not produce a soluble form that retains the catalytic function [57]. The host TMPRSS2 cleaves the S2 protein at the S1/S2 interface (the S2′ site) [43,51,133]. In addition to furin/furin-like proteases and TMPRSS2, other proteases may also be involved in the SARS-CoV-2 entry process, such as serine endoprotease proprotein convertase 1 (PC1), trypsin, matriptase (trypsin-like integral membrane serine peptidase), and cathepsins [51,100,128,129,130,134,135,136]. Proteases facilitate the infectivity of SARS-CoV-2 [51,137]. 

Labeling for TMPRSS2 is visible in the brush border (microvilli) of differentiated enterocytes in ileum, duodenum, jejunum, caecum, and colon [109,110]. TMPRSS2 is expressed in organoids composed of enterocytes, and ACE2 and TMPRSS2 are minimally expressed in blood cells [104]. TMPRSS2 is also observed within the respiratory epithelium [43]. TMPRSS2 is highly expressed on microvilli of the sustentacular cells, localized around the olfactory bulbs [111,120]. TMPRSS2 expression exhibits a gradient from upper to lower airways. TMPRSS2 is observed not only over microvilli membranes but also within the thin layer of the apical cytoplasm in ciliated cells and in only microvilli-containing cell [18,35,110,111,112,113,114]. However, it is not clear how TMDRSS2 is transported. Most likely, TMDRSS2 is localized in apical vacuoles and moves there with the help of actin comets.

## 7. Entry of SARS-CoV-2 into Cells

SARS-CoV-2 uses both clathrin-dependent and clathrin-independent endocytosis for cell entry [9,33]. The first step directed towards the entry of SARS-CoV-2 into cells lining human airways is its attachment to the plasma membrane (PM) with the help of ACE2. Cilia are an important mechanism promoting SARS-CoV-2 delivery to the cell bodies [34,35]. It was shown in a humanized in vitro model that the SARS-CoV-2 virus preferentially enters the tissues via ciliated cell precursors [36].

The binding of ACE2 to the S-protein induces endocytosis of the virion, after which the viral envelope fuses with the endosomal membrane to enable the release of the viral genome into the cytoplasm. However, direct fusion between membrane of the virion and the PM of the host cell cannot be excluded [51,95,101,138,139,140,141,142]. Alternatively, membrane fusion can also occur at the plasma membrane after receptor engagement, but only in cell cultures [51]. In vitro, viral particles are often observed along the filopodial membranes [10].

The necessity of an acidic environment for the processing of the S-protein also suggests the importance of endocytosis for SARS-CoV-2 entry into cells. Inhibition of TMPRSS2 activity blocks SARS-CoV-2 entry [15,17,18,43,143,144]. Rab5 is important for the delivery of SARS-CoV-2 to the early endosomes [145]. In pancreatic cells, ACE2, but not NRP1 and TMPRSS2, mediates SARS-CoV-2 entry [146]. Also, the endocytic cysteine proteases cathepsins B and L can cleave the subunits of the S-protein [130]. After cleavage of these subunits with proteases (including furin), the S-protein becomes more capable of “piercing” biological membranes [147,148]. Similarly, inhibitors of furin block entry of this virus into cells [32]. This suggests an endocytosis-dependent mechanism of SARS-CoV-2 entry. In furin-over-expressing cells, the expression of icSARS-CoV-2-GFP is higher than in wild-type cells. Labeling for TNPPSS2 was not observed on microvilli of the apical PM of sustentacular cells. This labeling was in dots below the APM (see Figure 3C presented in the paper by Khan et al. [120]). Enhanced expression of TMPRSS2 stimulates SARS-CoV-2 infection [120].

Of interest, the ^611^LY^612^ mutation impairs the glycosylation pattern of the S-protein and reduces its density on the surface of SARS-CoV-2. Mutations of cysteine-rich clusters I and II, the main palmitoylation sites, disrupt ER-to-Golgi transport of S-protein and reduce spike-mediated membrane fusion activity [47,149,150]. SARS-CoV-2-positive patients have higher expressions of ACE2, although the numerical density of TMPRSS2, BSG/CD147, and CTSB is lower than in patients with negative test for SARS-CoV-2 [34,48,97,126,138].

There are rare data suggesting that a low-pH environment is not a crucial determinant for the entry of SARS-CoV-2, similar to the previous observations on SARS-CoV, MERS-CoV and mouse hepatitis virus (MHV) [51,151]. Also, it was proposed that SARS-CoV-2 could enter cells directly without the participation of endocytosis [35]. According to this hypothesis, SARS-CoV-2 attaches to the cell surface in an ACE2-dependent manner and then, after cleavage of the S-protein, the membrane of the virus fuses with the APM, and the nucleocapsid enters the cytosol. From Figure 6C, presented by Pinto et al. [35], the authors conclude that the SARS-CoV-2 virion fuses to the plasma membrane. However, the so-called “fused virion” presented in this figure does not have enough peplomers (spikes) to prove that this protrusion is not a microvillus of the PM, but the real virion after its fusion with the PM. 

Even if the virus fused with the APM of the microvillus, there would be no space for the nucleocapsid to exit, and there are no mechanisms that would deliver the nucleocapsid components first to the apical part of the cell and after passing through the network composed of actin (and this is quite difficult if there are no special mechanisms) to the main body of the cell. On the contrary, if the virus is loaded via clathrin-dependent endocytosis into the apical endosome, then it has an actin comet that helps it pass through actin networks using actin and myosin located there or in the comet itself. 

After cutting off a part of S-protein, it becomes able to pierce the double membrane composed of the viral membrane and the part of the endosomal membrane to which the virus attaches. Calcium inhibits pore formation. It is present in the lumen of the endosome, but it is not present in the lumen of the airways. The membranes merge during bilayer dehydration with the help of polyethylene glycol. In order for the protein to be processed with TMPRSS2, it is necessary that the protein itself find it on the PM, which can be a very rare and random phenomenon.

Also, it is important to understand that if SARS-CoV-2 fused with the APM of microvilli, there is no space for the nucleocapsid or machine for the delivery of the nucleocapsid into the cell body. The basal part of the APM contains clathrin-coated buds, and immediately after the binding of the S-protein to ACE2, this complex would be captured into already existing clathrin-coated buds. Thus, the second pathways could be important only for nonpolarized cells in culture. Therefore, according to the current consensus, the endosomal pathway has a preference. Buds coated with clathrin are on the basis of the APM, from which the microvilli extend. These buds are located between the microvilli. However, it is not established whether clathrin-dependent vesicles formed from clathrin-coated bud fuse with early endosomes or not [51,140].

## 8. Alterations in Cells Induced by SARS-CoV-2

SARS-CoV-2 induces fragmentation of the Golgi complex (GC), down-regulates GRASP55, and up-regulates TGN46 expression, while the expression of GRASP55 or the knockdown of TGN46 reduces the infection rate. Lipid metabolism is altered [1,2,3,4,5,6,7,8,9]. The number of mitochondria accumulated at the periphery of the replication organelle decreases. The cortical actin is accumulated near the plasma membrane of the infected cells [13,152]. The SARS-CoV-2-dependent ER stress induces mitochondria fusion. The microtubules are not necessary for the biogenesis of SARS-CoV-2 or other coronaviruses [14]. SARS-CoV-2 triggers Golgi fragmentation via the down-regulation of GRASP55 [153].

SARS-CoV-2 induces overexpression of several genes controlling the ER stress, i.e., glucose-regulated protein 78 (GRP78 or BiP) and glucose-regulated protein 94 (GRP94) [14]. More than 300 host proteins binding to the SARS-CoV-2 RNA during active infection were identified [153,154]. Inhibition of fatty-acid metabolism or VPS34 blocks SARS-CoV-2 replication [154,155]. SARS-2 inhibits methylation machinery from host RNAs [156,157], SNAREs, and autophagy, preventing the capture of the double membrane vacuoles (DMVs) formed by the virus into autophagosomes. [158]. Although the mechanisms of intracellular transport are under revision [159,160,161], the role of intracellular transport in SARS-CoV-2 infection is important.

In the infected cells, the expression of ERGIC53 was slightly reduced. The GC appeared as small fragments dispersed in the cytoplasm. The spike was highly enriched in the Golgi fragments. These fragments did not contain stacks of cisterna but appeared as aggregates of convoluted tubules filled with virions (Figure 3P presented by Zhang et al. [153]).

Thapsigargin (a SERCA calcium pump inhibitor), tunicamycin (a protein glycosylation inhibitor), and dithiothreitol (a reducing agent); brefeldin A (an inhibitor of ArfGEF) and monensin (an inhibitor of trans-Golgi transport); bafilomycin A1 (an inhibitor of vacuolar-type H^+^-ATPase), chloroquine (which increases the pH in endosomes), vacuolin-1 (an inducer of large and swollen lysosomes), E64d, leupeptin, and pepstatin (inhibitors of lysosomal hydrolases); or a cocktail of protease inhibitors can significantly reduce SARS-CoV-2 infection. Both autophagy inducers (torin-1) and autophagy inhibitors (3-methyladenine), as well as CID 1067700, displayed similar inhibitory effects [153]. Treatment of cells with latrunculin A, which blocks actin polymerization, or with nocodazole, which induces depolymerization of microtubules, did not affect SARS-CoV-2 replication, whereas vinblastine inhibiting the assembly of microtubules had a strong effect on the production of infectious extracellular virions [13]. Thus, the inhibition of intracellular transport and endocytosis blocks SARS-CoV-2 infection. This suggests that without these protein machines, SARS-CoV-2 infection would be stopped. 

SARS-CoV-2 induces acute respiratory distress syndrome, which is characterized by alveolar epithelial necrosis at an early disease stage. Epithelial necrosis markers and in particular high-mobility group box-1 (HMGB-1) are released into the blood. Serum level of HMGB-1 is one of the damage-associated molecular markers released from necrotic cells. Alveolar epithelial cell necrosis involves two types of programmed necrosis, namely, necroptosis and pyroptosis [162,163]. 

COVID-19 infection leads to the disassembly of the Golgi ribbon and the mobilization of host cell compartments and protein machineries that are known to promote Golgi-independent trafficking to the cell surface [153,164]. Mutation of the C488 residue of the S-protein impairs its delivery at the GC and PM [165]. Brefeldin A blocks the processing of the S-protein [165].

In SARS-CoV-infected cells, mitochondria are larger and appeared as being fused (see Figures 4B and 5B presented by Snijde et al. [166]). In SARS-CoV-2-infected cells, there is a significant induction of monocyte-associated chemokines such as CCL2 and CCL8 [101,167,168,169]. IL13, a cytokine associated with Th2-high asthma, inhibits ACE2 expression [18]. Growth factor receptor signaling inhibition prevents SARS-CoV-2 replication [170,171]. SARS-CoV-2 activates mostly type I interferon responses [67]. Additionally, cells use the complex consisting of Drosha, Dicer, and RISC to degrade newly formed viral dsRNA. Drosha, Dicer, and RISC family proteins are conserved core components of the RNAi (RNA interference) machinery and are involved in post-transcriptional as well as transcriptional gene silencing in many eukaryotes. These machineries are saturable, and therefore, if the production of dsRNA were high, the cells would have problems due to accumulation of dsRNA in the cytosol and nucleus [172]. 

In order to protect dsRNA from degradation by RISC, SARS-CoV-2 uses DMVc with pores which prevent the movement of cytosolic proteins inside the lumen of DMVs. The viral dsRNA is accumulated in the DMV lumen [93,173]. SARS-CoV-2 disrupts splicing, translation, and protein trafficking to suppress host defenses. The papain-like protease of SARS-CoV-2 destroys p53 and inhibits interferon production [67,174,175,176,177,178,179].

Also, SARS-CoV-2 inhibits autophagy and prevents glycolysis, which is important for autophagy [14,180,181,182]. In any case, transfection of cells with LC3 protects the cells from SARS-CoV-2 infection [175,176,177,178,179,180,181,182,183]. However, SARS-CoV-2 can infect cells lacking LC3, ATG5, and ATG7.

## 9. Viral Replication Organelle

SARS-CoV-2 replicates its genome in the cytoplasm using host ribosomes for the synthesis of its own proteins. The positive-sense genomic RNA is considered by host ribosomes as mRNA. A mega-polypeptide is synthesized at the ER on the basis of the SARS-CoV-2 genome [33,158]. Then, SARS-CoV-2 forms viral replication organelle (VRO). It is composed of the paired convoluted reticular network of the ER membranes or the zippered ER (ZER). Also, numerous DMVs are formed. They are interconnected or connected with the ER. The ZER is found also in cells infected with Alpha-, Beta-, Delta-, and Gamma-coronaviruses [14,32,156,158]. The viral replication organelle is necessary for the initiation of viral protein synthesis. The negative-strand RNA functions as a template for multiple rounds of positive-strand RNA synthesis. The ZER exhibits uniform invaginations known as spherules. Spherules are comprised of a double membrane. The ZER has no luminal space, and it is connected with the rough ER membranes. Ribosomes are not associated with the ZER [13,14,24,166,184].

Another characteristic organelle which is formed by SARS-CoV-2 and by other coronaviruses is the DMV. DMVs are composed of double-paired membranes without space between single membranes. This is especially evident when quick freezing is used for analysis. Isolated DMVs can fuse with late endosomes/lysosomes. This indicates that DMVs contain SNAREs, at least STX17 and SNAP29. DMVs are situated near the ZER. In infected cells, DMVs appear rather quickly (within 2 h) after infection, and no indication that the amount of SER is responsible for the synthesis of FFAs is presented. The diameter of DMVs ranges from 160 to 400 nm. Ribosomes were occasionally detected on the cytosolic side of DMVs. In cells infected with SARS-CoV-2, abundant perinuclear DMVs are present, with an average diameter of 257 ± 63 nm [8]. Coronavirus replicase proteins are mostly detected in the ZER but not in DMVs. Aggregates of small-sized DMVs (diameter 185 nm ± 28 nm) are observed already at 6 h after infection. Then, their number and diameter (298 nm ± 42 nm) increase. Thin tubules connect outer membranes of some (not all) large DMVs with the ER. Small DMVs (19%) are mostly isolated [113]. The space between membranes of a DMV is empty and narrow. After quick freezing this space is ZER is surrounded with DMVs (Figure 1A of [185]). 

Replication takes place in connection with the replication organelle consisting of DMVs and the ZER. In SARS-CoV-infected cells, DMVs are sites of vRNA synthesis [166]. The lumen of DMVs does not contain cytosolic proteins but instead is filled with molecules of double-stranded form RNA (dsRNA). Single-stranded RNAs are not observed [7]. The length of individual dsRNA filaments in DMVs varies from 4–263 nm, with an average length of 52 nm [23]. 

In other coronaviruses, dsRNA inside DMVs was detected with antibodies against dsRNA and electron cryomicroscopy [13,66,166,185]. The presence of helicases in the viral genome suggests an important role of dsRNAs for the biogenesis of SARS-CoV-2. Helicases are required for the unwinding or despiralization of the dsRNA helix and thus for the efficient replication of +RNA viral genomes. The three-dimensional structure of the helicase–polymerase coupling in the SARS-CoV-2 replication–transcription complex is resolved [186]. 

We hypothesized that dsRNA serves as a matrix for the synthesis of new viral RNA genomes, whereas pores in the DMV walls formed by coronaviruses are necessary for protection of viral dsRNA [187] because in the cytosol, there are many RNases that can easily destroy single-strand RNAs.

## 10. Mechanisms of DMV Formation

When (+) RNA enters the cytosol of a cell, it, being composed of a single RNA chain, it is immediately attacked by cytoplasmic RNases. There should be mechanisms to protect it. One of these mechanisms is the synthesis of RNAs consisting of two chains that are packaged in DMVs. In order to have enough membranes for the treatment of DMVs within 4 h after the start of infection, a very rapid synthesis of fatty acids is necessary. However, it is very slow. Proteins that accelerate or stimulate the synthesis of additional lipids by the cell itself have not yet been found in the viral genome. Also, the molecular machines responsible for the synthesis of lipids are not found within viral proteins. Mitochondria-derived multilamellar organelles (MDMLOs) formed during mitochondrial fusion are better suited to explain such a rapid formation of DMVs. There is very little cholesterol in the OMM membrane and a lot of unsaturated LC; therefore, NSP6 with its 6 functioning TMDs, after being cut off from a giant polypeptide, can quickly penetrate into the lipid bilayer of the ER membrane [7].

DMVs are important for virion maturation and the protection of dsRNA from cytosolic RNases. If one assumes that DMVs are formed directly from the ER, several unknown mechanisms should be explained, namely, a mechanism for elimination of the ER matrix and mechanisms for the membrane attachment because NSP6 does not form dimers (see below). Until now, nobody has explained how the ER forms paired membranes, how NSP proteins and other viral proteins lacking signal peptides are inserted into the lipid bilayer, how the ER matrix is eliminated from the paired membrane domain, and how dsRNA is accumulated inside the DMV lumen (intermembrane space remains completely closed) using special pores similar to nuclear pores that apparently regulate the influx of proteins into the vacuole space [38,77,156].

As we indicated above, being transfected alone with NSP6 induces closure of the ER lumen with the formation of double paired membranes (DPMs) not containing lumen between two very tightly attached membranes [14,188]. This phenomenon cannot be explained by known mechanisms. Known mechanisms proposed for DMV biogenesis (see Figure 5 presented by van der Hoeven et al. [183]) cannot explain these problems [14]. Indeed, no evidence that paired ER domains formed by arteriviruses wrap around to form DMVs has been found [163]. Membranes of DMVs contain low levels of cholesterol [189]. Ribosomes were occasionally detected on the cytosolic side of DMVs formed by other coronaviruses [190].

Another possibility is to assume that two sheets of paired membranes attach to each other, as we found, and that their edges gradually would be transformed into a pore where proteins of nuclear pores would be accumulated; then, two sheets of paired ER would start to detach from each other, and the cavity filling would be regulated by the function of this pore, which would allow the passage of only dsRNA, as it was described for DMVs. Indeed, in SARS-CoV-2-infected cells, double paired membranes are observed when two DMVs make contact [7,14,138,156,163,183,188,189]. Of interest, after HPF and after a combination of chemical and cryo-fixation, quick-freezing demonstrated that DMVs of the hepatitis C virus also have almost completely closed space between membranes [190,191,192].

Very thin tubular junctions observed between the ER and the DMV induced the conclusion that DMV is formed from the ER [183,192]. During SARS-CoV-2 biogenesis, dsRNAs are formed. Viral dsRNA has been detected inside DMVs. DMVs prevent the Dicer-dependent degradation of dsRNA [8,15,112]. DMVs are identified as the main compartment where vRNA synthesis occurs for coronaviruses. In DMVs, there is no protein matrix in the intermembrane space, and the membranes are tightly glued to each other [6,138,166]. The excess of OMM formed after the fusion of mitochondria could be used for the formation of DMVs [193,194].

The synthesis (if lipids of DMVs are synthesized) and transformation of the ER cisternae (if membranes of DMVs are formed from the ER) should be very fast, otherwise RNA would be cleaved by cell RNases. However, cells cannot synthesize lipids with such high speed, and the molecular mechanisms responsible for the possible “cleaning” of the ER membranes are unknown.

## 11. Function of NSP6 and Mystery of DMV Production

Transfection of NSP6 induces closure of the ER lumen [77,188]. Transfection of cells induces the formation of paired (without any space between membranes) membranes connected to the ER [77]. The protein transfection technique used by Ricciardi et al. [77] itself raises a number of questions. Ricciardi et al. [77] indirectly demonstrate that NSP6-containing ZER has a low concentration of cholesterol. NSP6 from several coronaviruses localizes to mitochondria [195,196,197]. The reason for this, apparently, is that the OMM originated from the outer membrane of the Gram-negative archaea. Indeed, it is not clear how the NSP6 protein with six functional TMDs and having no signal peptide is included in the lipid bilayer of the ER membrane; how the membranes draw closer and stabilize in this position (NSP6 proteins have no property of sticking to each other [77]); or how the protein matrix is removed from the ER lumen to allow the membranes to converge to the point where they are closer together than in gap junctions. Such a feature is found only in initial autophagic vacuoles [198]. In infected cells, NSP6 is formed directly from the mega-polypeptide. Therefore, it has no signal peptide-dependent mechanism. The insertion of such a protein directly from the cytosol into the cholesterol-containing ER bilayers is extremely difficult. The only membranes glued together in the cell are early autophagosomes [198].

It was demonstrated that more than 30% of eukaryotic genes encode for signal polypeptides, which ensure their targeting to the ER membrane. Signal polypeptides are necessary for the insertion of such proteins into membranes, especially those containing cholesterol. Signal polypeptides represent specialized and indispensable targeting mechanisms that lead them to the ER membrane. For instance, mitochondrial porin is a major integral membrane protein of OMM. After its synthesis, an authentic yeast porin molecule is integrated into OMM. This porin does not interact with the ER membrane [199]. However, NSP6 has no signal peptide, which helps to introduce proteins with several TMDs into the bilayer. It is important to stress that in really infected cells, NSP6 is formed from a long peptide by cutting off a part of this peptide. In experiments in vitro, such a situation is rarely used. In the majority of experimental studies, NSP6 is transfected. In conditions of viral infection, the NSP6 protein is formed from a giant polypeptide by cutting it out of the chain using the NSP5 peptidase. After excision, the polypeptide chain of the NSP turns out to have no signal peptide, and it is very difficult for it to be inserted into the lipid bilayer of the ER membranes, as suggested in the hypothesis published at the end of this work. One of the possibilities suitable for the insertion of just-cleaved NSP6 into the lipid bilayer could be an excess of OMM formed after the fusion of mitochondria which then detaches, generating mitochondria-derived multilamellar vesicles (MDMLO). Then, MDMLOs fuse with the ER and are used for the formation of DMVs [194]. This hypothesis deserves additional analysis.

## 12. Virion Budding and Intracellular Transport

Coronaviruses (including SARS-CoV-2) use the host-cell proteins [200,201,202]. Intracellular transport of SARS-CoV-2 per se and its components could be used for the verification of the models describing this process. SARS-CoV-2 uses the ER for synthesis and processing of viral proteins, whereas membranes of ER exit sites (ERES) are used for the formation of viral lipid envelopes. The assembly of virions occurs on the cytoplasmic side of the ERES elements. A certain temperature range is necessary for SARS-CoV-2 assembly [203,204,205]. 

Upon co-expression of the E- or M-proteins, the S-protein is re-localized at ERES. The C-terminal retrieval motif within the cytoplasmic tail of the S-protein is necessary for its M-mediated retention in ERGIC. The E-protein is also involved in the retention of the S-protein [150]. S-protein is accumulated at the luminal side of the ERGIC membrane and forms a separate cylindrical complex [7]. A KxHxx motif in the cytosolic tail of the spike weakly binds the COPß’ subunit of the COPI coatomer, which facilitates some recycling of the spike within the Golgi, while releasing the remainder to the cell surface [21]. Protein/RNA complexes with reduced disorder are formed [14,206]. These complexes are composed of N-proteins and RNA, allowing efficient packing of the unusually large viral RNA genome into the small virus particles. Expression of S-, M-, E-, or NSP3 viral proteins triggers Golgi alterations [153].

Membrane proteins of SARS-CoV-2 are heavily glycosylated, suggesting their exit out of the ER and delivery at the GC [14,33,138,207]. Glycosylation of viral proteins occurs within 4 h after synthesis. This indicates that formation of DMVs, which need a huge amount of lipids, occurred during this time. The GC is required for glycosylation of viral proteins [14,129]. The numeric density of other coronaviruses is higher at the trans-pole of the GC [208]. The capsids could be formed during budding or assembled somewhere in the cytosol and then delivered to the site of virus budding. The size of SARS-CoV-2 virion suggests against the vesicular and the diffusion models of intra-Golgi transport. On the other hand, higher numerical densities of virions in cisternal distension at the trans-side of the GC suggests against the cisterna maturation–progression model of intra-Golgi transport [25].

There are images suggesting that SARS-CoV-2 particles budded in the post-Golgi vacuoles. In the vacuoles of infected cells, SARS-CoV-2 viruses are identical to particles found outside the cell membrane, suggesting that mature and infectious SARS-CoV-2 particles were already produced in the vacuoles [209]. The budding is observed only in the vacuoles, but not on the PM [210]. It is stated that coronaviruses use lysosomes for their secretion [209,211]. However, most of cargoes pass through endosomes, which are often LAMP1-positive [28,212,213,214,215]. 

Finally, the post-Golgi carrier (vacuole) filled with viruses fuses with the PM, and the viruses are secreted. It is not clear whether secretion occurs directly through APM, which is covered with mucus, or whether initially viruses are delivered at the BLPM and then into the lumen of airways. It is not clear whether the post-Golgi vacuoles could fuse with the BLPM. Analysis of the SNARE distribution is necessary to answer this question. After the luminal secretion, SARS-CoV-2 binds to the microvilli of the respiratory tract and induces the formation of apically elongated and highly branched microvilli that help SARS-CoV-2 pass through mucus [104]. These long microvilli explain why people who have recovered from COVID-19 suffer from coughing with much excreted mucus for a long time. 

## 13. Conclusions, Unclear Questions, and Perspectives

We proposed the following scheme for SARS-CoV-2 biogenesis (Figure 1). This scheme is based on nondynamic observations and cannot explain many simple questions. Although significant progress was made in the study of SARS-CoV-2, mechanisms of interactions of the SARS-CoV-2 virus with cells and especially its transport mode contain many unclear issues. Moreover, there is an enormous number of details at the molecular level of the SARS-CoV-2 general plan, but there is no understanding of the pathogenesis of SARS-CoV-2. Why ACE2 is transported towards the APM, whereas no apical sorting signals for it are found, is unknown. If ACE2 is the apically directed protein, why in cell cultures can non-polarized cells be infected? What are the mechanisms involved in SARS-CoV-2 budding? How are immature virions transported at the GC? What is the mechanism of intra-Golgi transport of immature and mature virions?

Secretory vacuoles filled with matured virions are secreted through the apical plasma membrane (APM) or baso-lateral PM (BLPM) in epithelial cells of airways. How do viral particles move towards the blood, passing the BM? How are RNAs protected against cytosolic RNases? Why it is important for SARS-CoV-2 to form dsRNA? Does dsRNA form the +RNA of the virion, and if so, how? What is the mechanism preventing movement of cytosolic protein inside DNVs, and how do pores in the double membrane of DMVs function? What is the mechanism involved in the insertion of NSP6 into the ER membrane, if the ER is the source for the formation of DMVs? What are the mechanisms responsible for the clearance of the lumen of the ER from its matrix proteins? Without knowledge on the structure and molecular biology of all abovementioned proteins and mechanisms of intra-Golgi transport, it is rather difficult to solve the problems of COVID-19. 

It is still not clear how viruses penetrate into the blood. In bronchioles, dendritic cells were not identified at the EM level. The resolution of immunofluorescence microscopy is not high, and cryo-sections have not yet been used. Importantly, dendritic cells are not found in alveoli. Molecular mapping, immunofluorescence, and immune-electron microscopy are necessary for the specification of the above issues related to the interaction between SARS-CoV-2 and cells. Thus, in spite of a huge number of publications on SARS-CoV-2, there remain many unclear questions within this field. They should be additionally examined from the point of view of intracellular transport.

## Figures and Tables

**Figure 1 ijms-24-04523-f001:**
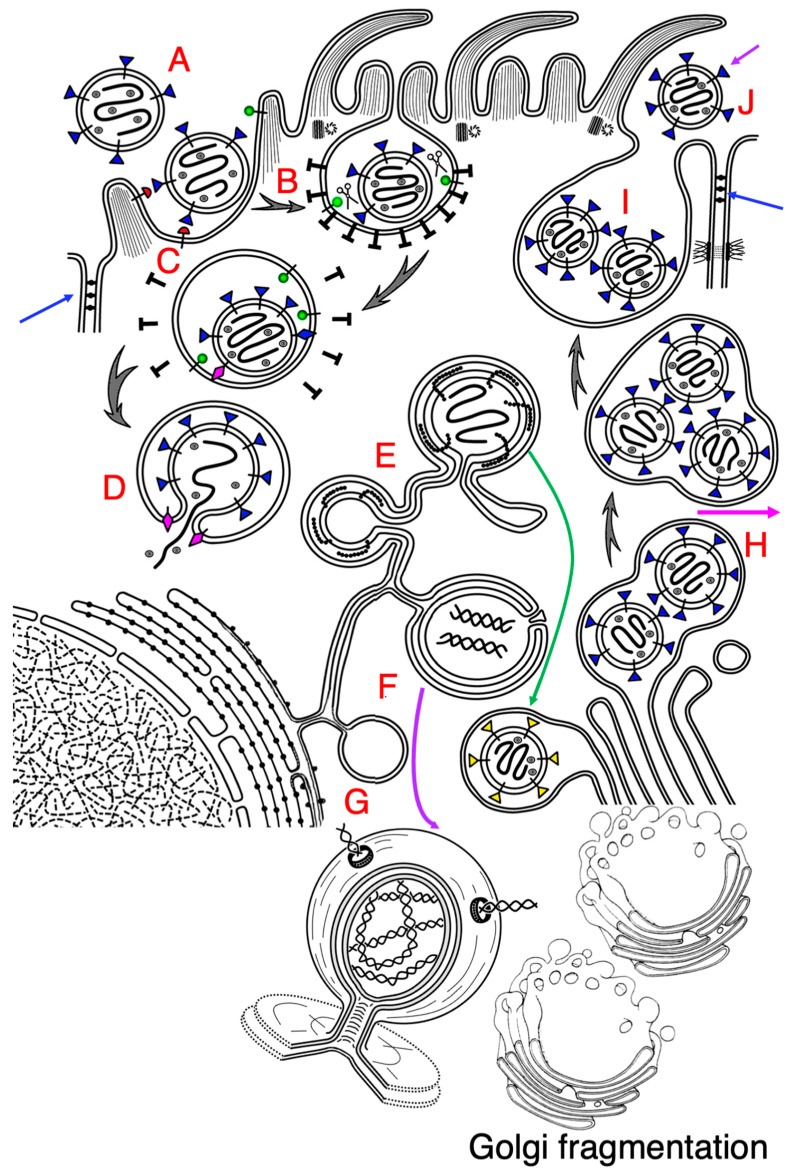
Biogenesis of SARS-CoV-2. The scheme shows interactions of SARS-CoV-2 with an epithelial ciliated cell. (**A**) After attachment of the virion to ACE2 (transmembrane structure with a hemispherical red head). (**B**) Clathrin-dependent endocytosis indices invagination of the apical plasma membrane and formation of endosome (**C**). Then, S-protein of SARS-CoV-2 (transmembrane structures with the triangle blue heads) is subjected to cleavage with TMDRSS2 (transmembrane structure with a spherical green head). It (magenta rhomb) perforates the endosomal limiting membranes (**D**). RNA of SARS-CoV-2 enters the cytosol (**D**) and forms the zippered endoplasmic reticulum (**E**). Simultaneously, double membrane vacuoles (DMV; (**F**,**G**)) are formed. (**G**) DMVs are filled with dsRNA and contain pores, through which dsRNA can move. Budding of the virus occurs within the ER exit site and ZER (**E**). During budding, the S-protein appears to be not fully glycosylated. After their budding, the viral particles are delivered to the Golgi complex and transported through it (green arrow). The immature (without glycosylation) virus passes through the GC, where it is subjected to high level of glycosylation. Viral particles are enriched (their numerical density increases) in the post-Golgi compartment (**H**). After concentration of viruses at the trans-side of the Golgi complex, the post-Golgi vacuole is delivered to the apical plasma membrane and fuses with it (**I**), and it secretes viruses into the lumen of airway ((**J**); magenta arrow) or into the space between epithelial cells (violet arrow). Blue arrows indicate tight junctions. SARS-CoV-2 induces fragmentation of the Golgi complex (see below to the right).

## Data Availability

Not applicable because the paper is the opinion based on the analysis of the piublished literature.

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
