# Peer review of "COVID-19 Biogenesis and Intracellular Transport"

_ijms, 2023, doi:10.3390/ijms24054523_

Round 1
Reviewer 1 Report
This manuscript is an interesting approach to the current knowledge about the intracellular biogenesis of coronaviruses (SARS-COVID in particular) and how it is processed and transported from and to plasma membrane along the endocytic and secretory pathways. The author has collected a lot of information and tried to be focus on biogenesis and trafficking of information, but in part this aim has not been gotten. There are parts that are nothing to do with this idea, which results at some moments very tedious to read and follow. I suggest the following to make the review more easily digestive for no expertise people. 1. There are repetitive information inside of the chapters. Read them again carefully and avoid this redundant information. Examples: lines 29-31 and lines 40-41; line 194 and then again line 199. 2. There are some confusing information in some parts or not understandable at all. It is necessary a more careful writing and revision of English. Some examples: 2.1. In page 103, author confuses “ciliated cells” with those containing microvilli” when talking of the digestive track. This misconception is repeated in pag 107. Therefore, I understand that microvilli containing cells are susceptible to be infected by coronavirus but not ciliated cells. 2.2. lines 57-58 and 59 are not clear at all. 2.3. in many parts the explanations of cells or abbreviations are indicated later on than they are referenced for the first time. Examples: line 133 for sustentacular cells are explained only in lines 176 and 181. APM (I guess that apical plasma membrane) is not defined anywhere. Some explanation is necessary for understanding what organoid cells are (lines 168 and 169). I would say “organoids” instead of organoid cells. ZER (line 264) and later on its definition (line 267). 2.4. If ZER has no luminal space, why it is represented in Fig. 1? This figure must be more precise and consistent with the text. The legend is not easy to follow either. 3. There is information inside some chapters that are nothing to do with it. Examples: Line 206 and on is nothing to do with ACE2 localization. 4. Chapters 7 and 8 are little to do (for not saying “nothing to do” with the aim of the manuscript. 5. I suggest moving chapter 9 to the beginning of the review and/or to elaborate a Table containing the reported molecular components indicated in this chapter. 6. There are some typographic errors beginning from the own abstract. Fig. 2 indicated at the end of the review should be Figure 1 (lines 724-725). 7. I suggest including at the end of the review a list of unresolved questions regarding the biogenesis and trafficking of coronavirus of SARS-COVID disease. In sum, the review is of interest for requires extensive changes to facilitate its huge containing information as well as to reorder chapters and simplify and order some information inside of each chapter. It is also necessary to draw a more accurate Figure 1 scheme accordingly to what is said in the text.Author Response
To editor
We rewrote the text almost completely, eliminated excessive copying of texts taken from other papers, added a list of unclear questions, changed the consequence of chapters, prepared Table 1 and a new Figure 1, changed Figure 2, checked and corrected English. New texts are colored in yellow. Old texts are colored in green.
Reviewer 1
This manuscript is an interesting approach to the current knowledge about the intracellular biogenesis of coronaviruses (SARS-COVID in particular) and how it is processed and transported from and to plasma membrane along the endocytic and secretory pathways. The author has collected a lot of information and tried to be focus on biogenesis and trafficking of information, but in part this aim has not been gotten. There are parts that are nothing to do with this idea, which results at some moments very tedious to read and follow.
Our reply: Thanks a lot for these words.
Reviewer 1
I suggest the following to make the review more easily digestive for no expertise people.
1a. There are repetitive information inside of the chapters.
Our reply:
We re-wrote the text trying to avoid to use excessive amount of the texts from other papers
Reviewer 1
1b. Read them again carefully and avoid this redundant information. Examples: lines 29-31 and lines 40-41; line 194 and then again line 199.
Our reply:
We corrected these mistakes and completely re-wrote the text.
Reviewer 1
- There are some confusing information in some parts or not understandable at all. It is necessary a more careful writing and revision of English.
Our reply:
We corrected our English
Reviewer 1
Some examples: 2.1. In page 103, author confuses “ciliated cells” with those containing microvilli” when talking of the digestive track. This misconception is repeated in pag 107. Therefore, I understand that microvilli containing cells are susceptible to be infected by coronavirus but not ciliated cells.
Our reply:
We corrected these mistakes and completely re-wrote the text. Ciliated cells have microvilli (see new image). From the papers published it is not completely clear whether only microvillar cells are infected or also ciliary cells are infected through microvilli. We stressed this problem in the text.
Reviewer 1
2.2. lines 57-58 and 59 are not clear at all. 2.3. in many parts the explanations of cells or abbreviations are indicated later on than they are referenced for the first time. Examples: line 133 for sustentacular cells are explained only in lines 176 and 181. APM (I guess that apical plasma membrane) is not defined anywhere.
Reply
We corrected this.
Reviewer 1
2.3. Some explanation is necessary for understanding what organoid cells are (lines 168 and 169). I would say “organoids” instead of organoid cells. ZER (line 264) and later on its definition (line 267).
Reply
We corrected this.
Reviewer 1
2.4. If ZER has no luminal space, why it is represented in Fig. 1? This figure must be more precise and consistent with the text. The legend is not easy to follow either.
Reply
We corrected this.
Reviewer 1
- There is information inside some chapters that are nothing to do with it. Examples: Line 206 and on is nothing to do with ACE2 localization.
Reply
We corrected this.
Reviewer 1
- Chapters 7 and 8 are little to do (for not saying “nothing to do” with the aim of the manuscript.
Our reply:
We changed the title of the paper and included some information from these chapters into other chapters.
Reviewer 1
- I suggest moving chapter 9 to the beginning of the review and/or to elaborate a Table containing the reported molecular components indicated in this chapter.
Reply
We corrected this
Reviewer 1
- There are some typographic errors beginning from the own abstract. Fig. 2 indicated at the end of the review should be Figure 1 (lines 724-725).
Reviewer 1
- I suggest including at the end of the review a list of unresolved questions regarding the biogenesis and trafficking of coronavirus of SARS-COVID disease. In sum, the review is of interest for requires extensive changes to facilitate its huge containing information as well as to reorder chapters and simplify and order some information inside of each chapter. It is also necessary to draw a more accurate Figure 1 scheme accordingly to what is said in the text.
Reply
We corrected this.
Reviewer 2 Report
In this review, the author summarizes a broad spectrum of the life cycle of the SARS-CoV-2 virus, the causative of COVID-19 disease, covering viral entry, assembly, trafficking, egress, spike processing, ACE2 localization, cell defense responses, and so on. The intention of this review is good by including as many aspects as possible for the non-virologists. However, this is also the flaw of this manuscript of covering so many topics that the contents are not well organized and do not have a clear logical flow, which makes it not friendly for the readers. Moreover, there are too many obvious spelling and grammar mistakes that must be corrected, and improvements in writing should be done before acceptance to publish in the International Journal of Molecular Sciences. It will be interesting to cell biologists if the author makes more efforts to organize and improve the writing. So, I suggest this manuscript for major revision.
Specific points:
1. If the review can be focused on the intracellular transport of SARS-CoV-2, it will be much better than the current version. Some parts of the review can be deleted, like Proteins of SARS-CoV-2.
2. Some sentences show up twice which is very unnecessary and confusing. For example, line 500-500 and line 505-506; line 606 and line 617.
3. Spelling and grammar of the manuscript must be improved. For example, in line 7, hrough should be through; in line 8, SARS-CoV2 should be SARS-CoV-2; in line 10, the past tense and current tense appear in the same sentence; in line 23, COVID189 should be COVID-19; in line 38, SARS-o should be SARS-CoV; in line 40 and many other places, SARS-CoV-1 was used, in one paper, either SARS-CoV or SARS-CoV-1 should be used, but not at the same time; line 57-59, we do not analyze…were not analyzed should be corrected; in line 84, both ACE-2 and ACE2 were used; in line 187, enter should be enters; in line 200, in higher should be is higher; in line 208, result should be results; in lines 378 and 379, ORF8 should be NSP8; …etc.
Author Response
Reviewer 2
In this review, the author summarizes a broad spectrum of the life cycle of the SARS-CoV-2 virus, the causative of COVID-19 disease, covering viral entry, assembly, trafficking, egress, spike processing, ACE2 localization, cell defense responses, and so on. The intention of this review is good by including as many aspects as possible for the non-virologists. However, this is also the flaw of this manuscript of covering so many topics that the contents are not well organized and do not have a clear logical flow, which makes it not friendly for the readers. Moreover, there are too many obvious spelling and grammar mistakes that must be corrected, and improvements in writing should be done before acceptance to publish in the International Journal of Molecular Sciences. It will be interesting to cell biologists if the author makes more efforts to organize and improve the writing. So, I suggest this manuscript for major revision.
Reply
We corrected fulfilled all demands
Specific points:
- If the review can be focused on the intracellular transport of SARS-CoV-2, it will be much better than the current version. Some parts of the review can be deleted, like Proteins of SARS-CoV-2.
Our reply: We rewrote the text almost completely, eliminated excessive copying of texts taken from other papers, added a list of unclear questions, changed the consequence of chapters, prepared Table 1 and a new Figure, checked English.
- Some sentences show up twice which is very unnecessary and confusing. For example, line 500-500 and line 505-506; line 606 and line 617.
Our reply: We corrected this.
- Spelling and grammar of the manuscript must be improved. For example, in line 7, hrough should be through; in line 8, SARS-CoV2 should be SARS-CoV-2; in line 10, the past tense and current tense appear in the same sentence; in line 23, COVID189 should be COVID-19; in line 38, SARS-o should be SARS-CoV; in line 40 and many other places, SARS-CoV-1 was used, in one paper, either SARS-CoV or SARS-CoV-1 should be used, but not at the same time; line 57-59, we do not analyze…were not analyzed should be corrected; in line 84, both ACE-2 and ACE2 were used; in line 187, enter should be enters; in line 200, in higher should be is higher; in line 208, result should be results; in lines 378 and 379, ORF8 should be NSP8; …etc.
Our reply: We corrected all these and other mistakes.
Reply
See our reply to reviewer 2.
Round 2
Reviewer 1 Report
Tha review has significantly improved in organization and figures.
Author Response
Reviewer 1
Thе review has significantly improved in organization and figures.
Reply
Thanks a lot
Reviewer 2 Report
In this revised review, the author re-organized the context and corrected many grammar and spelling mistakes. Overall, it has be much improved. However, there are still some concerns that need to be addressed before acceptance to publish in the International Journal of Molecular Sciences.
Specific points:
1. In Figure 2, there are lipid bilayers for the plasma membrane, Golgi apparatus, and viral particles, but only one layer of lipid is shown for ER and nucleus, which looks very weird and must be modified.
2. Golgi fragmentation is one of the most dramatic morphological alterations in infected cells, but the Golgi seems intact in Figure 2.
3. Figure 1 and table 1 overlayed. It will be better to remove Figure 1, because table 1 includes most information of figure 1.
4. In line 24-25, there are 7 members of coronaviruses, which can infect human beings, or there are 7 members of coronaviruses, which induces dangerous diseases.
5. In line 26, the spike number on each SARS-CoV-2 viral paricle varies from different literature. So, it should be a range, about 25-40.
6. There are still many grammar and spelling mistakes. For example, in line 66, passe should be pass; in line 68, contains should be contain; in line 78, after “transmembrane domain” should be a ,; in line 83-83, the sentence needs to be modified; in line 196, “ACE2 are” should be ACE2 is; in the main text, both ACE2 and ACE-2 are used, which needs to keep consistent; line 340-342, a citation should be included; etc.
Author Response
Reviewer 2
In this revised review, the author re-organized the context and corrected many grammar and spelling mistakes. Overall, it has be much improved. However, there are still some concerns that need to be addressed before acceptance to publish in the International Journal of Molecular Sciences.
Specific points:
- In Figure 2, there are lipid bilayers for the plasma membrane, Golgi apparatus, and viral particles, but only one layer of lipid is shown for ER and nucleus, which looks very weird and must be modified.
Reply
We corrected this and added new information.
- Golgi fragmentation is one of the most dramatic morphological alterations in infected cells, but the Golgi seems intact in Figure 2.
Reply
We corrected this and shown Golgi fragmentation.
- Figure 1 and table 1 overlayed. It will be better to remove Figure 1, because table 1 includes most information of figure 1.
Reply
We prefer to keep this Table in the text because there are a lot of references necessary for readers. The Figure 1 we included into the graphical abstract.
- In line 24-25, there are 7 members of coronaviruses, which can infect human beings, or there are 7 members of coronaviruses, which induces dangerous diseases.
- In line 26, the spike number on each SARS-CoV-2 viral paricle varies from different literature. So, it should be a range, about 25-40.
- There are still many grammar and spelling mistakes. For example, in line 66, passe should be pass; in line 68, contains should be contain; in line 78, after “transmembrane domain” should be a ,; in line 83-83, the sentence needs to be modified; in line 196, “ACE2 are” should be ACE2 is; in the main text, both ACE2 and ACE-2 are used, which needs to keep consistent; line 340-342, a citation should be included; etc.
We fulfilled all these demands, corrected all mistakes and additionally checked our English.